# Investigation of the Microcharacteristics of Asphalt Mastics under Dry–Wet and Freeze–Thaw Cycles in a Coastal Salt Environment

**DOI:** 10.3390/ma12162627

**Published:** 2019-08-18

**Authors:** Qinling Zhang, Zhiyi Huang

**Affiliations:** 1College of Civil Engineering and Architecture, Zhejiang University, Hangzhou 310000, China; 2College of Water Conservancy and Architecture Engineering, Tarim University, Alaer 843300, China

**Keywords:** asphalt mastics, coastal salt environment, microcharacteristics, dry–wet and freeze–thaw cycles, correlation analysis

## Abstract

In the coastal areas of southeastern China, high temperatures and humidity in the summer and microfreezing in the winter, as well as a high concentration of salt spray in the environment, seriously deteriorate the durability of asphalt mixtures. Therefore, the microcharacteristics of asphalt mastics (asphalt mixed with mineral filler) under the effect of chlorine salt and “dry–wet and freeze–thaw” (DW-FT) cycles were investigated by Fourier-transform infrared (FTIR) spectroscopy, gel permeation chromatography (GPC), and atomic force microscopy (AFM) techniques. Two factors, including asphalt mastic types (base and styrene-butadiene-styrene (SBS)-modified mastics) and numbers of DW-FT cycles, were considered based on the natural environment. Regression functions were established to explore the relationship between the FTIR, GPC, and AFM indexes. The results indicate that there were no chemical reactions between the asphalt and filler because the infrared spectrum of the base and SBS-modified mastics were similar. With the increase of the salt “DW-FT” cycle numbers, the sulfoxide index and large molecular size ratios (LMS%) increased, and the surface roughness ($R_{q}$ and $R_{a}$) of the morphology decreased, as illustrated by a flatting mastics surface phenomenon in the AFM test. Regression analysis confirmed that there was a high correlation between the FTIR, GPC, and AFM indexes, and formation of the bee structures was closely related to the long chain index. The SBS-modified mastics had a better antiaging performance with a lower increase in the sulfoxide index after the salt “DW-FT” cycles in the coastal environment.

## 1. Introduction

More and more asphalt pavement and sea-crossing bridges are being paved with asphalt material in southern coastal regions in China. The harsh external environmental conditions (high salt, high temperature, and humidity in the summer and micro-freeze–thaw in the winter) on asphalt pavement results in serious impacts. Approximately 77.75% of seawater salt is sodium chloride [1]. Therefore, the performance of asphalt mixtures is simultaneously affected by water and chloride salt [2]. The salt accumulated on the surface and inside of the asphalt pavement aggravates the aging of the asphalt materials due to the erosion or crystallization of chloride ions after dehydration [3,4]. Obika et al. [5] conducted research about the damages produced by the salt that reaches the bituminous mixtures in hot climates, determining that the salt crystallization, solubility, and crystal pressures have important roles in the damage produced in the mixture. Luis et al. [6] researched the mechanical properties of three types of bituminous mixtures affected by salt (sodium chloride, NaCl), and they observed that the hot mix asphalt was scarcely affected when it was submerged in salt water and was also unaffected when salt was added as aggregate. However, the porous mixture is more susceptible to the effects of salt for all salt interactions, especially when the aggregate has been submerged in salt water before making the bituminous mixture. The salt in the porous mixtures has a harmful effect: it causes a loss of internal friction. Starck et al. [7] insisted that the de-icing agent decreased the stiffness and increased the softening point of the binder, and the viscosities of de-icing agent binders had risen after immersion in de-icing agent solutions. This indicated the binder aged during the freeze–thaw (FT) cycles and salt solution soaking process. In certain areas, pavement undergoes many cycles of dry–wet (DW) and freeze–thaw (FT) in a single year. Currently, many researchers have focused on the effects of de-icers and snow melting agents on asphalt materials, such as sodium and calcium chlorides on macrocharacteristics of asphalt concrete (strength, low temperature crack resistance, fatigue life, and others), especially in the United States, Canada, Japan, and China [8,9,10]. Shi et al. [11,12] also demonstrated the damage effects of chloride salt with a de-icer solution on the performance of asphalt pavements and found a series of water damages (loosening and pits in the asphalt pavement occurred). In addition, some researchers have studied the damage of sulfate to asphalt pavement in saline soil areas in China. Xiong et al. [13,14] selected DW cycles or FT cycles in a sulfate solution to accelerate the corrosion of an asphalt mixture, and they found that the brucite fiber in the mixture has an excellent effect on stability, enhancement, and reinforcement and, thus, improves the durability of the asphalt mixture. Regarding the microstructure, Cui et al. [15,16] studied the influence of FT cycles in a salt solution on three routine indicators of the rheological property and microstructure of asphalt binder by SEM, and the analysis results showed that the high- and low-temperature performances of the binders were reduced before and after FT cycles. However, empirical test methods utilized physical parameters (penetration, viscosity, softening point, and ductility) that measured only colligative properties and did not fully access the fundamental engineering properties of asphalt materials [17].

Presently, chemical techniques (Fourier-transform infrared (FTIR) spectroscopy and gel permeation chromatography (GPC)) can offer specific information related to molecular composition, including functional groups and molecular size [18,19]. In addition, the atomic force microscopy (AFM) technique is being used more widely in the study of asphalt materials to examine their nanosized structures and nanomechanics [20,21,22,23,24,25]. However, little work has been done on the micro-characteristics of mastics exposed to coastal salt environments by FTIP, GPC, and AFM techniques, which may provide more information about chemical structures and morphological relationships of asphalt material.

Therefore, this paper aims to examine the actions of salt “DW-FT” cycles on the chemical structures and micromorphologies of the mastics in the southern coastal environment in China. Two types of mastics (base and styrene-butadiene-styrene (SBS)-modified mastics) were prepared, and the salt “DW-FT” cycle test was designed based on actual conditions. Functional group indexes were obtained by the FTIR test. The weight-average molecular weight ($M_{w}$), the number-average molecular weight ($M_{n}$), z-average molecular weight ($M_{z}$), and large molecular size ratio (LMS%) indexes were obtained by the GPC test, and the AFM test was adapted to characterize changes in morphological properties of the asphalt mastics sample surface. Regression statistics were conducted on these indexes to explore the relationships between the chemical structure and morphology of the asphalt mastics.

## 2. Materials and Methods

### 2.1. Material Properties

The base asphalt binder (AH-70) and SBS-modified asphalt binder (SBS-I-D) were used in this paper, which were generally applied on the asphalt pavement in southern coastal regions in China and were supplied by Sinopec Zhenhai Refining & Chemical Company (Ningbo, China). Their physical properties are presented in Table 1 and met China’s technical specification requirements (JTG E20-2 011) [26].

Limestone fillers were produced by Jingmen (Hubei Province, China) in this paper. Every technical indicator met China’s technical specification requirements (JTG F40-2004) [27], and the results are shown in Table 2.

The chlorine salt (NaCl) used in this paper was analytically pure sodium chloride, and its technical indicators were in accordance with China’s standard specifications (GB/T 1266-2006).

### 2.2. Experimental Work

#### 2.2.1. Preparation of Asphalt Mastics

A constant filler/asphalt (by mass) ratio of 1.0 was selected in accordance with Superpave specifications, which recommend a ratio within 0.6–1.2 [28]. Asphalt mastics were prepared following the optimized protocol to obtain homogeneous asphalt-filler mixes from a previous investigation [29]. The prepared mastics were poured into a flat, stainless-steel tray and placed in an oven at 120 °C for 5 min to form a <10 mm thick and uniform, thin layer.

The short-term aging method was conducted using the thin film oven test (TFOT). The mastics were aged for 5 h at 163 °C in air in accordance with China standards (JTG E20-2011) [26]. The aged mastic samples were used for subsequent salt “DW-FT” cycle tests.

#### 2.2.2. Salt “Dry–Wet and Freeze–Thaw (DW-FT)” Cycles Test

The salt “DW-FT” cycles test is first used in civil engineering to evaluate the impact of “DW-FT” cycles on the performance of cement concrete [30,31] and soil [32,33]. In terms of the asphalt concrete “DW-FT” cycles test, there are no standards because of the differences in climate and environmental conditions. In reference to previous research [34,35,36], a modified “DW-FT” cycles test was proposed in this paper.

Combined with the climate and environmental characteristics of the coastal region of Ningbo, and referenced in previous literature [37,38], the NaCl solution concentration was 0% (pure water) and 5% by weight of water. Before the salt “DW-FT” cycles test, the samples were soaked in a water/salt solution for 2 h. Then, the samples were removed and placed in an oven for 30 min at 60 °C, followed by immersion in the water/salt solution for 30 min at 25 °C. These two steps were defined as one DW cycle. After 8, 15, and 25 successive DW cycles, the mastic samples were subjected to 8 successive FT cycles. One FT cycle consisted of freezing in a water/salt solution at −10 °C for 30 min and thawing in air at 15 °C for 30 min. Therefore, there were four types of salt “DW-FT” cycles (0–0 cycles, 8–8 cycles, 15–8 cycles, and 25–8 cycles). The outline of the salt “DW-FT” cycle test plan is displayed in Figure 1.

The salt “DW-FT” cycles test was conducted in a high-/low-temperature environmental test chamber, which included an automatic temperature and humidity controller. This machine reduced the risk of any human error.

#### 2.2.3. Fourier-Transform Infrared (FTIR) Test

The FTIR spectra were measured by the AVA TAR370 FTIR of the American NICOLET company, which is available in the Analysis Center of the Agrobiology and Environmental Sciences, Zhejiang University. The spectra were recorded from 4000 to 400 cm^−1^ at a resolution of 4 cm^−1^ by averaging 32 results for each measurement. The FTIR spectral bands were processed and analyzed by OMNIC 8.0 software included in the testing instrument and Origin 2017 software. A KBr pellet technique was used, where 10~0.1 mg of each sample was mixed with 1000–0.1 mg of KBr in an agate mortar [39]. 

#### 2.2.4. Gel Permeation Chromatography (GPC) Test

Waters GPC equipment (Waters 1525/2414, Figure 2) with computerized data acquisition software was used in this study and was available as a test platform in the State Key Laboratory of Chemical Engineering (Zhejiang University, Hangzhou, China). During testing, the columns were kept at a constant temperature of 40 °C in a column oven, and the mobile phase was a tetrahydrofuran (THF) at a flow rate of 1.0 mL/min.

The samples were prepared following the procedure described by Moraes et al. [40]. First, approximately 20 mg of the mastic samples were dissolved in THF in a 10 mL volumetric flask for 30 min while hand shaking was applied (Figure 3). Second, the solution was filtered through a weighted 0.45 and 0.25 μm syringe filter, respectively. Then, 50 μL of the dissolved sample was injected into the GPC injector module for each test. One test took 40 min, and elution started approximately 17 min after injection and ended at approximately 28 min. Each sample was tested in three replicates, and the average value was calculated. 

#### 2.2.5. Atomic Force Microscopy (AFM) Test

In this study, a Bruker dimension icon AFM (Figure 4), which is available in the Analysis Center of Agrobiology and Environmental Sciences (Zhejiang University, Hangzhou, China), was applied to investigate the topography of the mastics at room temperature (20 °C), and the humility was approximately 85%. For each sample, an area of 20 × 20 μm (512 × 512 pixels) was scanned at a rate of 1 Hz in the peak force tapping mode. A cantilever with a spring constant of 0.4 N/m was adapted. To ensure the repeatability of the observations, three points were observed for each sample.

The AFM observation sample of the mastics was produced by the hot drop method [22,41,42]. First, the base and SBS-modified asphalt mastics were heated at 120 °C and 135 °C, respectively, and then approximately 20 mg of each of the hot liquid mastics were coated on glass slides (approximately 10 × 10 mm, as shown in Figure 5. Then, the mastic-coated slides were placed back in the oven for 5 min to obtain a smooth surface for scanning. Finally, the samples were stored inside a desiccator at room temperature at least for 24 h before AFM tests.

## 3. Results and Discussion 

### 3.1. FTIR Test Results

#### 3.1.1. FTIR Characteristics of the Asphalt Binder, Filler, and Mastic

The spectra for the binder, mineral filler, and mastic after a short-term aging process are shown in Figure 6; the x-axis is the wavenumber (cm^−1^), and the y-axis is the absorbance.

As seen in Figure 6, for the spectra of the asphalt binder, filler, and mastic, there was a large and broad band found near 3419 cm^−1^, which was ascribed to the hydroxyl (O−H) vibrations [43].

Compared to the infrared spectrum of the base binder, the spectrum of the SBS-modified binder included not only all the peaks of the base binder but also special peaks at 967 cm^−1^, which correspond to the butadiene double bonds (−CH=CH−). It can be seen in Figure 6 that the peak spectrum characteristics were similar, and the absorption intensity was slightly different. Please note that no distinct peak was observed at 1700 cm^−1^, which corresponds to the C=O bond for the base and SBS-modified binder, indicating that their carbonyl was zero and that slight thermal aging took place in the TFOT-aged process. This result agrees with previous research [44,45].

Spectral bands of filler occurred at wavenumbers 2509, 2512, 1797, 1427, 876, and 712 cm^−1^ and corresponded to the characteristic peaks of the carbonate compounds, which are the main components of limestone mineral fillers [43].

The functional groups of the base and SBS-modified mastics were the combination of functional groups of the base binder, SBS-modified binder, and mineral filler. No new peaks appeared, which indicated that the fusion between fillers and binders was mainly a physical adsorption, and no chemical reaction occurred.

A detailed analysis of the absorption peaks of asphalt binders, fillers, and mastics is shown in Table 3.

#### 3.1.2. Changes in the Spectra of the Mastics 

FTIR spectroscopy was expected to show some of the structural changes that occurred during the salt “DW-FT” cycles test. FTIR spectra of all mastics are shown in Figure 7. As shown in Figure 7, similar spectra containing similar peaks and valleys were obtained before and after the salt “DW-FT” cycles, which indicates that no new functional groups were introduced. The only difference was in the intensities of the peaks. This indicated that the test conditions changed the content of certain functional groups of the mastics. 

As seen in Figure 6 and 7, the overlapping bands of the mastics in the ranges of 1780 to 900 and 900 to 600 cm^−1^ were more serious and covered the functional group and fingerprint areas.

#### 3.1.3. Semiquantitative Analysis

To obtain a semiquantitative estimate describing the variation of the main functional groups, the overlapping areas of the bands from deconvolved spectra were analyzed, and the ratios of selected absorption bands were calculated. But, the overlapping bands made it difficult to directly evaluate parameters such as position, width, and area [46]. To resolve this problem, a curve-fitting technique, which is mainly based on the Levenberg–Marquardt algorithm [47,48], was proposed as a useful and robust method to estimate these band parameters because of its potential to resolve overlapping bands into distinct peaks [49,50,51]. Many studies have also validated the curve-fitting method in medical science [52,53,54] and optical engineering applications [55]. Therefore, curve-fitting was used to identify functional groups and estimate the numbers, positions, intensities, and widths of the subpeaks in the overlapping bands. 

Spectra data were pre-processed by OMNIC 8.0 software using spectra averaging, smoothing, and baseline correction. The positions and numbers of the bands were established initially from the second derivative of the spectrum using the OMNIC 8.0 software, and the shapes, heights, and widths of the overlapping bands were performed by curve-fitting using Origin 2017 software. The curve-fitting process was in the wavenumbers ranging from 1780–900 cm^−1^ and 900–600 cm^−1^ of the base mastics, as an example, to illustrate the feasibility of the curve-fitting method for the qualitative and semiquantitative study of the mastics (Figure 8). 

As seen in Figure 8, the red line is the second derivative of the measured spectrum, the green line is the fitted subpeak, and the black line and yellow line are the measured spectra and the synthesized spectra, respectively. As indicated by the vertical lines, all subpeaks were identified by the second derivative technique, which perfectly matched the resolved peaks. The curve-fitting of the overlapped peaks in the original spectrum was performed using the curve fit module of the Origin 2017 Software. In Figure 8, there were 4 subpeaks in the 900–600 cm^−1^ range and 10 subpeaks in the 1780–900 cm^−1^ range of the mastics’ spectra. Detailed wavenumber positions of the subpeaks are shown in Table 4 and Table 5, and the functional groups belonging to the subpeaks are identified (Table 3).

To semiquantitatively analyze the aging action of the salt “DW-FT” cycles on the mastics, different functional group indexes were calculated by considering the area of the functional bands.

The key functional group indexes are the aromatic index $A_{1600}/\sum^{\text{​}}A_{1200-600}$, the branched aliphatic index $A_{1377}/(\sum^{\text{​}}A_{1377}+A_{1460})$, the sulfoxide index $A_{1032}/\sum^{\text{​}}A_{1200-600}$, the long-chain alkane index $A_{723}/(\sum^{\text{​}}A_{1377}+A_{1460})$, and the butadiene index $A_{967}/\sum^{\text{​}}A_{1200-600}.$ The subscript represents the wavenumbers at which the absorbance was determined, and A was the area of the corresponding spectral band, which can be obtained using the Feak analysis module of the Origin 2017 software. The values of the branched aliphatic index and aromatic index can reflect the changes to the aliphatic structure and aromatic structure, respectively, while the values of the sulfoxide index and butadiene index can reflect changes to the oxygen-containing functional groups [56,57]. The long-chain alkane index is likely related to the solid wax phases in asphalt [58].

Variations of the functional group indexes of the base and SBS-modified mastics before and after the salt “DW-FT” cycles are shown in Table 6.

The sulfoxide index of the mastics shows an increasing trend. Further analysis indicates that the increasing rates in the sulfoxide indexes of the base mastics were 7.5%, 20.14%, and 46.75%, and for the SBS-modified mastics were 4.67%, 16.83%, and 40% after the salt “DW-FT” cycles. Therfore, the increased values of the SBS-modified mastics are slower than that of the base mastics. In other words, SBS-modified mastics experienced lower oxidation levels than the base mastics. Some studies have found (although some disagreement exists) that the use of SBS in the modification of asphalt improves the flexibility at low temperatures and thereby enhances the cold weather performance of the modified asphalt [59]. 

A remarkable decrease in the long-chain indexes and butadiene indexes was observed, which is an indication that the mastic molecules also suffered from chain scission during the salt “DW-FT” cycles. The aromatic index and branched aliphatic index irregularly changed after the salt “DW-FT” cycles, and this shows that the reaction between the components of the mastics during the salt “DW-FT” cycles is more complicated.

### 3.2. GPC Test Results

GPC is a size-exclusion chemistry technique that yields the molecular weight distribution of the analytes based on molecular size [60]. 

#### 3.2.1. Molecular Size

Figure 9 shows the GPC chromatograms for mastics before and after the salt “DW-FT” cycles. The chromatogram was evaluated between 17 and 28 min, which was the range in which the end of the mastic molecule elution began. For each chromatogram, the area under the curve represents 100% of the mastic molecules injected into the GPC system. The general trends of all the mastic curves were almost consistent, but there were visible differences, such as the peak size and the intensity, at some specific positions.

To better illustrate these molecular weight distribution structure changes, the range of the response distribution from the GPC test is usually divided into 13 slices of equal elution time areas [61], and these areas are then separated into three fractions: the first 5 slices are defined as large molecular size (LMS), the next 6–9 slices are defined as medium molecular size (MMS), and the rest of the area under the curve is referred to as small molecular size (SMS) [62,63]. The *LMS%*, *MMS%* and *SMS%* equations are provided in Equation (1) to Equation (3) [63,64,65,66,67] as follows: (1)$$LMS\%=\frac{Aare\;of\;middle\;\frac{5}{13}\;of\;chromatogram}{Total\;area\;beneath\;chromatogram}\times 100,$$
(2)$$MMS\%=\frac{Aare\;of\;middle\;\frac{4}{13}\;of\;chromatogram}{Total\;area\;beneath\;chromatogram}\times 100,$$
(3)$$SMS\%=\frac{Aare\;of\;last\;\frac{4}{13}\;of\;chromatogram}{Total\;area\;beneath\;chromatogram}\times 100.$$

Analysis of the GPC chromatograms and calculation of the *LMS%*, *MMS%* and *SMS%* were calculated utilizing Origin 2017 software. As shown in Figure 10, aged mastics had a higher content of LMS%. As the number of salt “DW-FT” cycles increased, the LMS% growth of the SBS-modified mastics was less than the LMS% growth of the base mastics. This conclusion was consistent with the results of the FTIR test.

#### 3.2.2. Molecular Weight 

The molecular weight parameters ($M_{n}$, $M_{w},$ and $M_{z}$) of all mastics before and after the salt “DW-FT” cycles were automatically measured using Empower 3 software with the GPC system. 

Equation (4) shows the aging indexes calculated from the molecular weight parameters [40]. Figure 11 depicts the calculated results for the mastics after the salt “DW-FT” cycles.
(4)$$\left(M_{n},\;M_{w},\;M_{z}\right)=\frac{\left(M_{n},\;M_{w},\;M_{z}\right)\;\mathrm{after}\;\mathrm{salt}\;“\mathrm{DW}-\mathrm{FT}”\;\mathrm{cycles}}{\left(M_{n},\;M_{w},\;M_{z}\;\right)\;\mathrm{before}\;\mathrm{salt}\;“\mathrm{DW}-\mathrm{FT}”\;\mathrm{cycles}}\;=\frac{{\left(M_{n},\;M_{w},\;M_{z}\right)}_{\left(8-8\right)\;\mathrm{or}\;\left(15-8\right)\;\mathrm{or}\;(25-8)}}{{\left(M_{n},\;M_{w},\;M_{z}\right)}_{\left(0-0\right)}}$$

As shown in Figure 11, the most significant difference between the aging indexes of the mastics was observed for the $M_{z}$ index (average molecular size corresponding to the *LMS%*), where $M_{z}$ dropped from 1.10 to 1.04 for the base mastics and from 1.05 to 1.03 for the SBS-modified mastics. Therefore, the aging index reduction rate of the SBS-modified mastics was lower than that of the base mastics. The GPC test suggests that there were a greater number of chemical species assemblies with different molecular weights and shapes in the mastics. These findings could be largely attributed to the polymerization reactions of smaller components (possibly involving waxes and asphaltenes, ~SMS) into LMS components (resins and asphaltenes) that occurred during the salt “DW-FT” cycles test.

### 3.3. AFM Analysis

#### 3.3.1. AFM Morphology

Topographic images (at a scale of 20 μm) of the mastics are shown in Figure 12. There are many randomly distributed bee structures, which are similar to the experimental results presented in the research findings [68]. With the addition of fillers, the three typical structures were in the dispersed phase (bee structures), matrix phase (dark region), and some luminous dots (mineral filler particles) appeared in the mastic topographic images. 

As shown in Figure 12a, the impression of the bee structures was formed from alternating higher and lower parts in the surface topography of the mastics. Nano-scope analysis software 1.40 of AFM was used to measure the system, randomly select the typical three structures in Figure 12a, and extract the height information in Figure 12b. In the direction of the arrow, the surface height of the bee structure (blue line) was in the form of a “wave”, the height of the matrix phase (green line) was relatively flat, and the height of the luminous dots (red line) was in the form of a mountain. 

Topographic images of all mastics before and after the salt “DW-FT” cycles are shown in Figure 13. Many bee structures were tightly surrounded with mineral fillers in the mastics. No obvious bee structures can be observed in Figure 13a_2_,a_4_ in the topographic images of the binder mastics, which may be hidden in the fillers. The bee structures became bigger in size and fewer in number after the salt “DW-FT” cycles in the SBS-modified mastics, as seen in Figure 13b, and this suggests that the effect of interactions between the water, salt and components of the mastics had already occurred, which lead to morphologic changes to the sample surface’s topography.

The surface height distribution frequencies of all mastic samples obtained by the depth analysis function of Nano-scope Analysis software 1.40 are shown in Figure 14. The value of the surface height with the highest frequency was selected as the representative value of the sample height. It can be seen from Figure 14 that the height distributions of the surface of all mastics were similar to the normal distribution. The test height of the mastics under different test conditions is summarized in Table 7.

As seen in conjunction with Figure 14 and Table 7, the surface height of the mastic sample before the salt “DW-FT” cycles was significantly larger than the surface height of the mastic sample after the salt “DW-FT” cycles. The surface height of the base mastics and SBS-modified asphalt mastic samples decreased with an increase in the salt “DW-FT” cycle numbers. In the salt environment, the microscopic morphologies of all mastics changed, which reduced the spatial difference of all mastics between the phase states on the nanometer scale. In addition, the surface of the microscopic phase transitions from a multiphase state to a single-phase state became flatter [69]. Therefore, the salt “DW-FT” cycles test changed the microscopic morphology of the mastics and lead to a decrease in the macroscopic properties of the mastics in the salt environment.

#### 3.3.2. Microstructural Quantification 

From the above microstructure evolution analysis on the nanometer scale, the surface height of the mastic samples had a significant change in the salt “DW-FT” cycles. The difference of the phase states reflected the performance between the mastic components. This paper selected the surface roughness parameters to quantify this difference of the phase states. The commonly used surface roughness parameters have a root-mean-square roughness parameter $R_{q}$ and an arithmetic mean roughness $R_{a}$ [69], and their calculation formulas are Equation (5) and (6), respectively.
(5)$$R_{q}=\sqrt{\sum_{i=1}^{N}Z_{i}^{2}/N},$$
(6)$$R_{a}=\frac{1}{N}\sum_{i=1}^{N}Z_{i}.$$

In the process of calculation, three parallel microscopic observation samples were selected for every sample, three test areas were selected for each observation sample, and 512 × 512 pixel points were selected for statistical analyses in each test area. The surface roughness parameters of all mastics can be automatically obtained by Nano-scope analysis 1.4. Figure 15 is the quantitative result of $R_{q}$ and $R_{a}$ of all mastics in the different test conditions.

As shown in Figure 15, the $R_{q}$ and $R_{a}$ gradually decreased with the increase in the salt “DW-FT” cycle numbers, and the nanoscale phase height difference gradually decreased. Further analysis indicated that the decreasing rates in $R_{q}$ of the base mastics were 25.05%, 60.11%, and 74.78%, and the SBS-modified mastics were 12.29%, 18.77%, and 30% after the salt “DW-FT” cycles. In addition, the decreasing rates in$\;R_{a}$ of the base mastics were 34.14%, 60.87%, and 80.69%, and the SBS-modified mastics were 5.97%, 12.23%, and 20.23%. The result of $R_{q}$ and $R_{a}$ showed that the trend of the microstructure transition from multiphase to a uniform-phase structure was, again, confirmed, which is consistent with the research results of the surface height distribution (Figure 14). Therefore, whether the surface height or the surface roughness index is used can be quantified to determine the reaction of the action of the salt environment. Because of the surface roughness parameters, the SBS-modified mastics decreased at a slower rate compared to the base mastics, which showed that the SBS-modified mastic had good aging resistance in the salt “DW-FT” cycles test.

### 3.4. Correlation between the Chemical Composition and Microstructure Properties

To investigate the correlation between the chemical composition and the microstructure surface, regression statistics for these correlations are shown in Figure 16 and Figure 17.

The relationships between the sulfoxide index, *LMS%*, and$\;R_{q}$ and $R_{a}$ are plotted in Figure 16. There was a strong correlation between the sulfoxide index and the $R_{q}$ and $R_{a}$, with *R*^2^ values of 0.80 and 0.79 of the base mastics and *R*^2^ values of 0.89 and 0.94 of the SBS-modified mastics, respectively. According to the results of Branthaver et al. [70], the sulfur content correlated with the asphaltene content. This was because there was a large amount of sulfur in the asphaltene fraction of the asphalt mastics. 

For the base mastics, there was a strong, linear relationship between the sulfoxide index and the *LMS%*, with an *R*^2^ of 0.78. However, for SBS-modified mastics, no strong relationships were found between the sulfoxide index and the *LMS%*, with a *R*^2^ of 0.42. This is mainly because the sulfur in the base mastics is easily oxidized, and the oxidation rate is faster, while the asphalt phase and the SBS modifier in the SBS-modified asphalt cross-link with each other, and the SBS modifier first degrades and inhibits the sulfur formation in the salt “DW-FT” cycles. This phenomenon also verified that the composition of the SBS-modified mastics at the molecular scale was rather complicated compared to the base mastics.

As shown in Figure 17, the long-chain index correlates well with *LMS%*, with an R^2^ of 0.95 for the base mastics and 0.98 for the SBS-modified mastics, and also correlates well with AFM roughness parameters $R_{q}$ and $R_{a}$, with *R*^2^ values of 0.96 and 0.99 of the base mastics and *R*^2^ values of 0.86 and 0.98 of the SBS-modified mastics, respectively. The result proves that the bee structure in AFM may have a strong relationship with the solid wax crystal based on the FTIR long-chain index in the mastics.

## 4. Conclusions

In this study, the effects of salt, moisture, and temperature in a coastal environment in southern China on the chemical composition and micro-nanostructure of base- and SBS-modified mastics were studied. The relationships between the chemical properties and the morphologies of the mastics were examined. The present study will be useful in furthering the understanding of the characteristics of mastics in the salt “DW-FT” cycles. The results can be summarized as follows:

(1) FTIR is an effective, semiquantitative tool to study the chemical properties of mastics. In this study, the FTIR test showed that the lower experimental temperature (60 °C, a high temperature for the asphalt pavement surface in the summer) exhibited little influence on the carbonyl (C=O) of asphalt mastics, but this temperature exhibited a bigger influence on sulfoxide (S=O). Therefore, the sulfoxide index can be used as the aging index in the salt “DW-FT” cycle test. 

(2) FTIR spectra show that there are no chemical reactions between mineral fillers and asphalt binders, and there are also no chemical reactions between the salt solution and mastics in the salt “DW-FT” cycles. The styrene-butadiene-styrene in SBS-modified mastics may improve aging resistance through minimal reduction in the sulfoxide index.

(3) The morphology of the AFM test clearly indicated that the bee structures increased in size and decreased in number, and the roughness of the morphology was gradually reduced, as illustrated by a flattening phenomenon of the mastics’ surface with increased salt “DW-FT” cycle numbers. 

(4) The FTIR indexes (the sulfoxide index and long-chain index), GPC index (*LMS%*, $M_{z}$), and AFM index $(R_{q}\;$and $R_{a}$) were usually used to characterize the aging characteristics of asphalt materials and showed different changing trends with increased salt “DW-FT” cycle numbers. Overall, there were good correlations between the FTIR indexes, GPC indexes, and AFM indexes. The aging phenomenon of the salt environment on asphalt material is similar to that caused by high-temperature oxygen aging, but the aging mechanism should be different and needs further research.

(5) The results of this paper also show that SBS-modified mastics exhibited better aging resistance to the salt environment compared to the base mastics. Therefore, SBS-modified mastics are more suitable in southern coastal regions in China.

It should be noted that the correlations presented in this paper are based on only two kinds of mastics. Therefore, further work is needed with a broader range of asphalt materials and different experiment methods to confirm the conclusions in this paper, and greater exploration of the mechanism between mastics and the salt environment is needed in future research.

## Figures and Tables

**Figure 1 materials-12-02627-f001:**
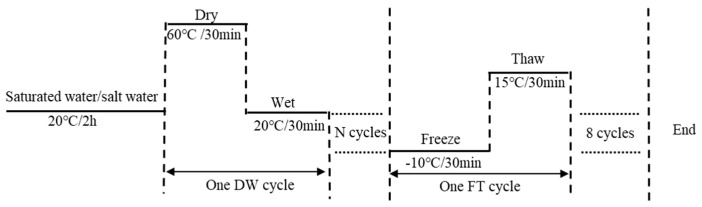
Schematic of the salt “dry–wet and freeze–thaw” (DW-FT) cycles for the mastics: N = 8, 15, and 25.

**Figure 2 materials-12-02627-f002:**
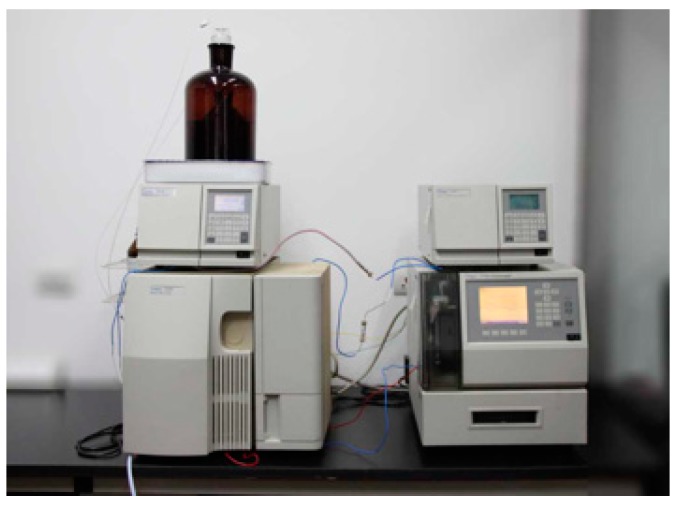
Gel permeation chromatography (GPC) system.

**Figure 3 materials-12-02627-f003:**
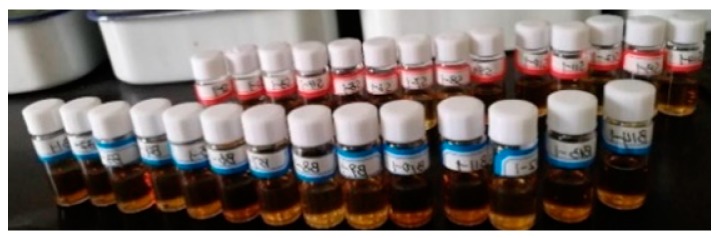
GPC samples.

**Figure 4 materials-12-02627-f004:**
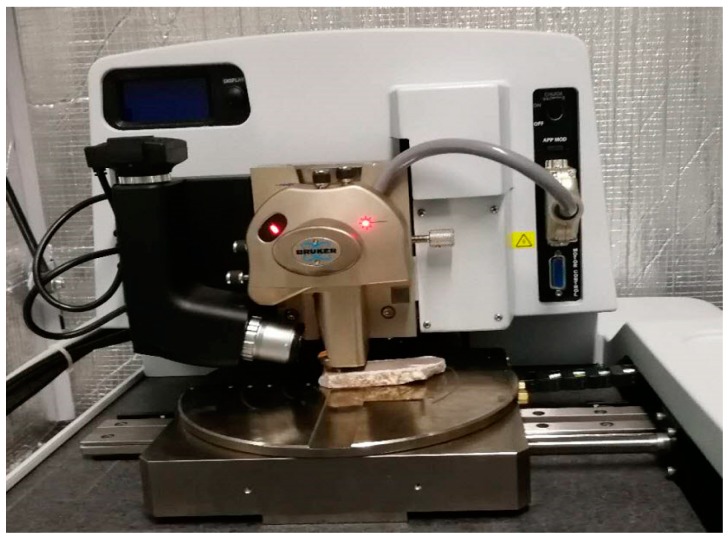
Atomic force microscopy (AFM) system.

**Figure 5 materials-12-02627-f005:**
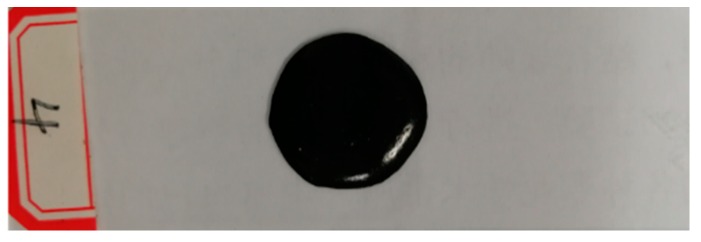
AFM samples.

**Figure 6 materials-12-02627-f006:**
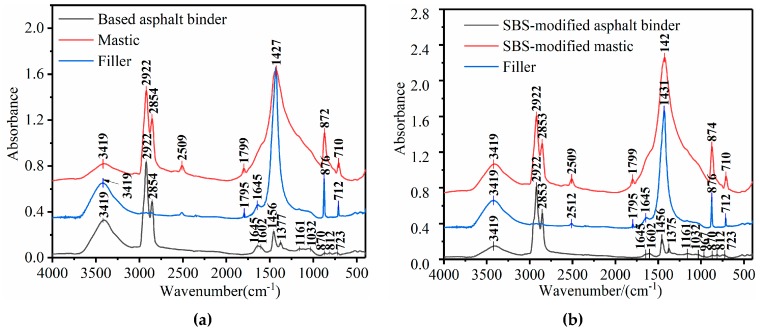
Spectra of asphalt binder, filler, and mastic. (**a**) Base mastic; (**b**) SBS-modified mastic.

**Figure 7 materials-12-02627-f007:**
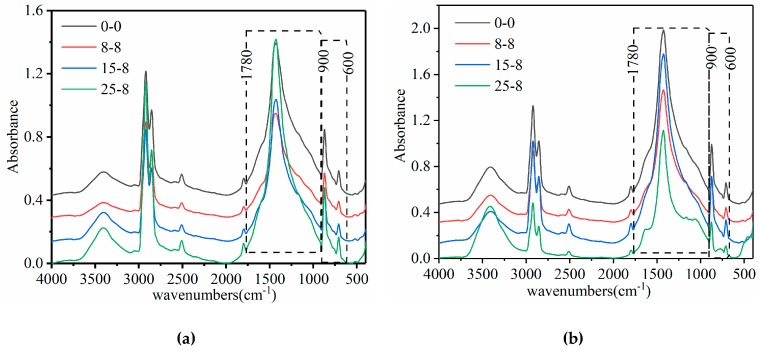
FTIR spectra of mastics in different test conditions. (**a**) Base mastics; (**b**) SBS-modified mastics.

**Figure 8 materials-12-02627-f008:**
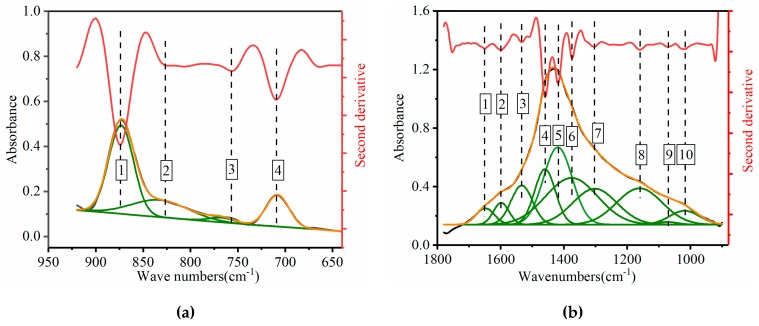
Graph showing the curve-fitting of the overlapping bands for the mastics. (**a**) 900–600 cm^−1^ wave numbers; (**b**) 1780–900 cm^−1^ wave numbers.

**Figure 9 materials-12-02627-f009:**
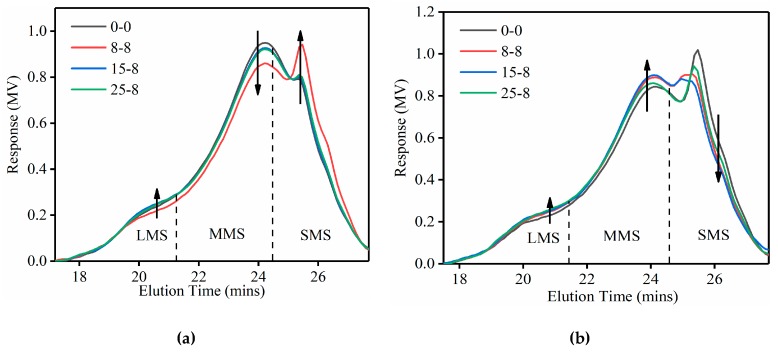
GPC curves of mastics in different test conditions. (**a**) Base mastics; (**b**) SBS-modified mastics. Note: LMS—large molecular size, MMS—medium molecular size, and SMS—small molecular size ratios.

**Figure 10 materials-12-02627-f010:**
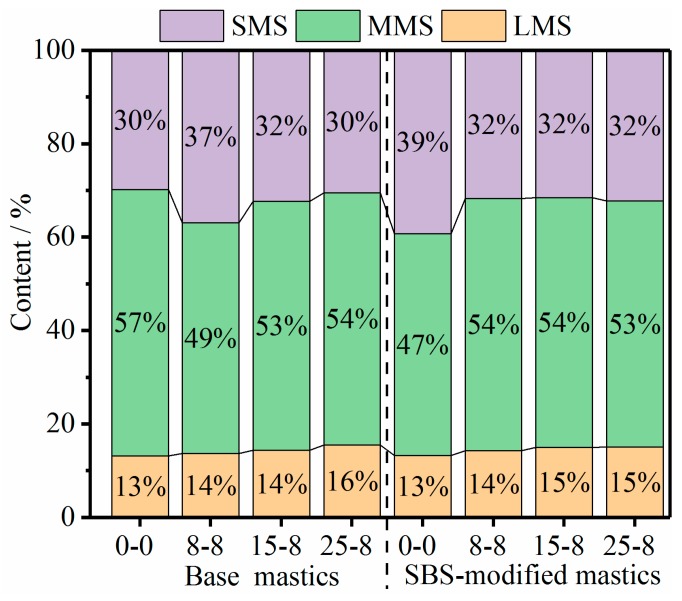
Large molecular size (LMS), medium molecular size (MMS), and small molecular size (SMS) ratio changes of the mastics.

**Figure 11 materials-12-02627-f011:**
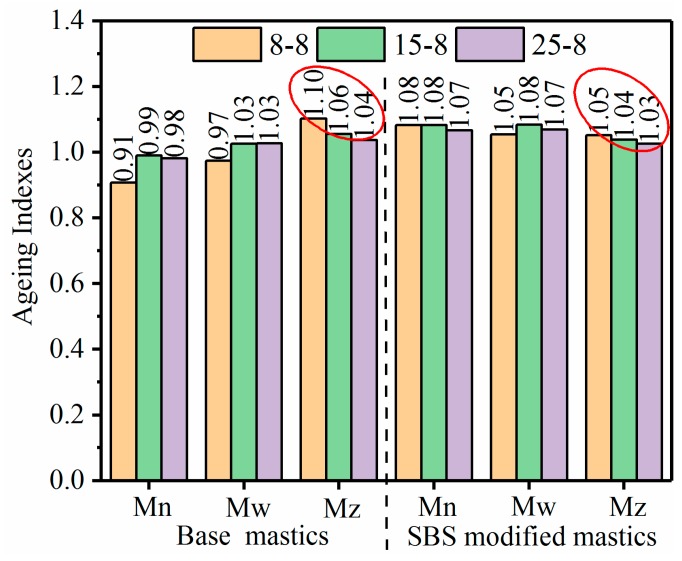
The ageing indexes of the mastics.

**Figure 12 materials-12-02627-f012:**
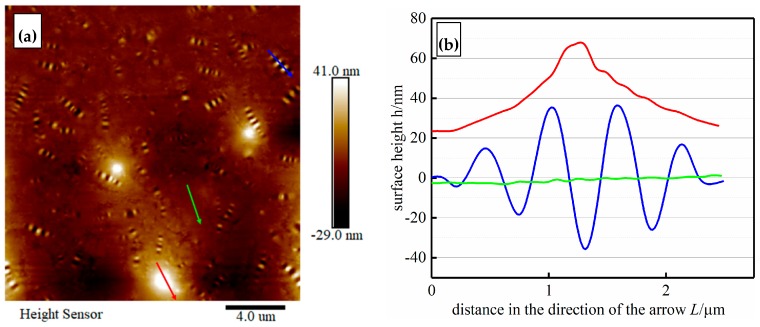
2D AFM morphology images (**a**) and surface height curves along arrow directions of the mastics (**b**).

**Figure 13 materials-12-02627-f013:**
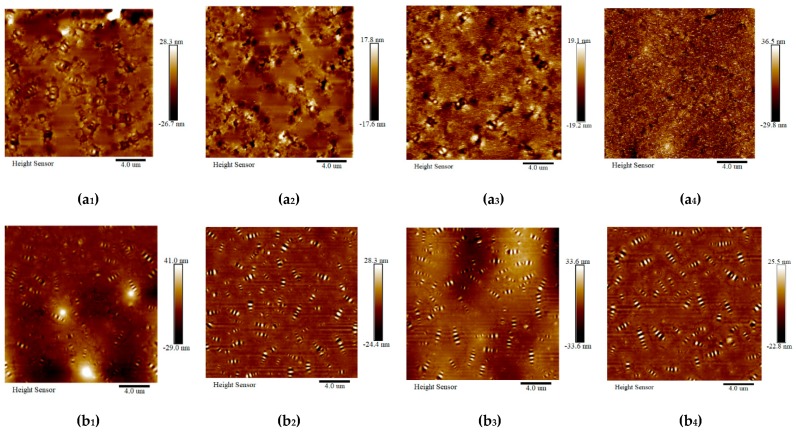
AFM topographic images of the mastics (20 × 20 μm). (**a_1_**) Base mastics (0–0); (**a_2_**) Base mastics (8–8); (**a_3_**) Base mastics (15–8); and (**a_4_**) Base mastics (25–8). (**b_1_**) SBS-modified mastics (0–0); (**b_2_**) SBS-modified mastics (8–8); (**b_3_**) SBS-modified mastics (15–8); and (**b_4_**) SBS-modified mastics (25–8).

**Figure 14 materials-12-02627-f014:**
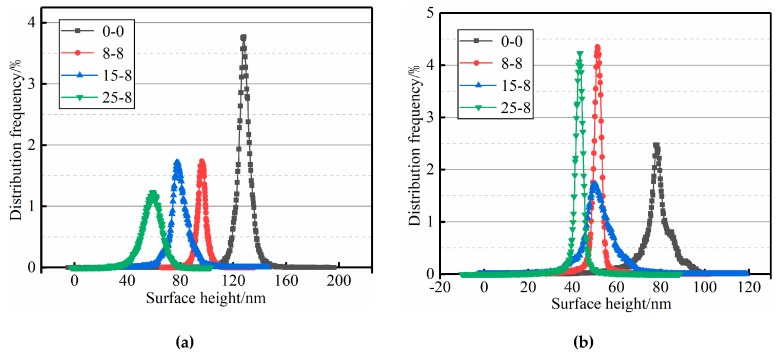
Distribution frequency of the mastic surface heights. (**a**) Base mastics; (**b**) SBS-modified mastics.

**Figure 15 materials-12-02627-f015:**
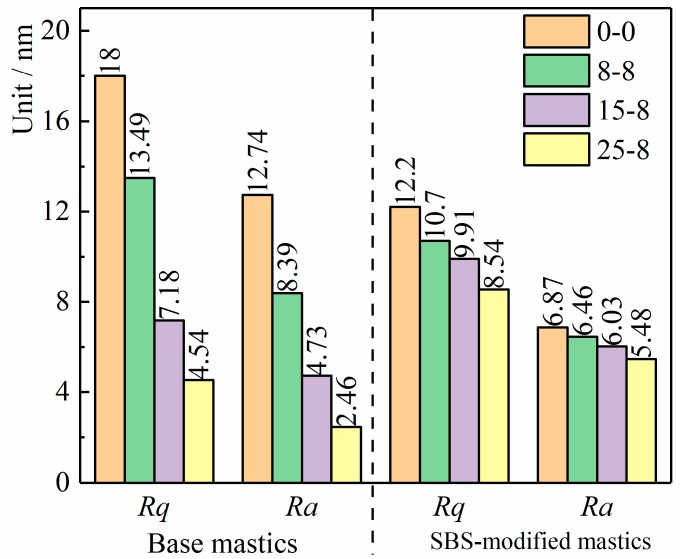
Roughness statistical analysis of the mastics.

**Figure 16 materials-12-02627-f016:**
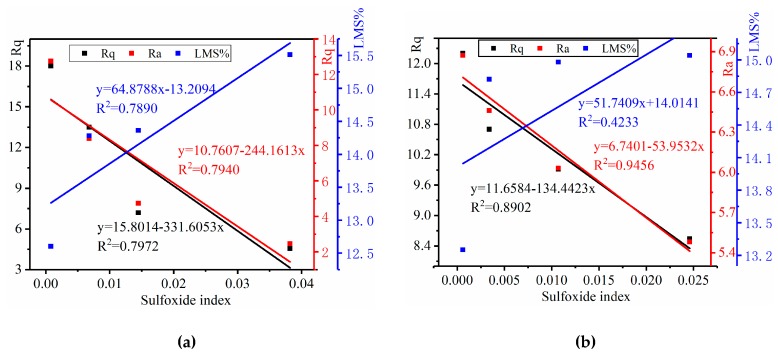
Relationship between the sulfoxide index, LMS%, and AFM index. (**a**) Base mastics; (**b**) SBS-modified mastics.

**Figure 17 materials-12-02627-f017:**
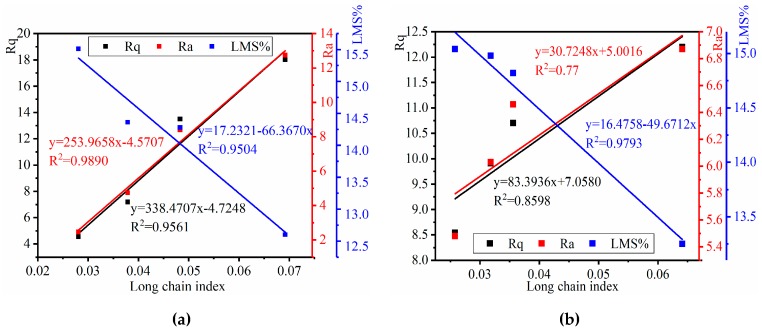
Relationship between the long-chain index, LMS%, and AFM roughness. (**a**) Base mastics; (**b**) SBS-modified mastics.

**Table 1 materials-12-02627-t001:** Properties of asphalt binders.

Properties		AH-70 base Binders		Styrene-Butadiene-Styrene (SBS)-I-D Modified Binders	
		Standard Values	Test Results	Standard Values	Test Results
Penetration (25 °C, 100 g, 5 s), 0.1 mm		60–80	74	40–60	55
Softening point, °C		44–57	47.3	≥60	86.6
Ductility (5 °C, 5 cm/min), cm		≥100	>150	≥20	31.5
Wax content, %		≯3.0	1.1	-	-
RTFOT	Mass loss, %	≤0.8	0.01	≤1	0.08
	Penetration ratio (25 °C), %	≥55	68.9	≥65	71.6
	Ductility (5 °C, 5 cm/min), cm	≥30	68.4	≥15	17.3

Note: 1. SBS-I-D:Polymer modified asphalt adopts China’s technical specification requirements (JTJ 036-98), and Class I is SBS thermoplastic rubber polymer modified asphalt. The I-D type is suitable for the heavy traffic sections in hot areas. 2. RTFOT: Rotating thin film oven test.

**Table 2 materials-12-02627-t002:** Properties of mineral fillers.

Properties		Test Results	Standard Values	Test Method
Apparent density, g/cm^3^		2.798	≥2.50	T0352
Water content, %		0.49	≤1	T0332
Through the percentage of sieves, %	<0.6 mm	100	100	T0351
	<0.15 mm	95.23	90~100	
	<0.075 mm	85.91	75~100	
Hydrophilic coefficient		0.78	<1	T0353

**Table 3 materials-12-02627-t003:** Peak position and explanation of Fourier-transform infrared (FTIR) spectra.

Wave Numbers (cm^−1^)	Explanation
3419/1645	Hydroxy (−OH) asymmetric stretch and angular vibration of water molecules
2922/2854/2853	Methy(−CH_2_−) symmetry vibration of the aliphatic long chain (saturated)
2509/2512	${(\mathrm{CO}}_{3}^{2-})$ combination frequency of antisymmetric and symmetric stretching vibrations
1795/1799	${(\mathrm{CO}}_{3}^{2-})$ weak symmetrical stretching vibration and in-plane bending vibration
1602	Asymmetric ring vibration of the benzene ring and carboxyl group
1460/1456/1431/1425/1427	Symmetric bending vibration of −CH_2_ groups and aliphatic long chains (saturated)
1377/1375	Methy l(−CH_3_) umbrella vibration and aliphatic branched chain (saturated)
1161	Aliphatic sulfonic acid (SO_2_) symmetric stretching
1032	Sulfoxide (S=O) stretching vibration
967	Butadiene (SBS) stretching vibration
876/712	${(\mathrm{CO}}_{3}^{2-})$ out-of-plane bending vibration and in-plane bending vibration
870/812	Stretching vibration of benzene ring
723	Covibration of the methylene segment (−CH_2_−)_n_ (n ≥ 4)

**Table 4 materials-12-02627-t004:** Curve-fitting wavenumbers of subpeaks of asphalt mastics in the 900–600 cm^−1^ range.

Numbers	1	2	3	4
Base and SBS-modified mastics	870	812	723	710

**Table 5 materials-12-02627-t005:** Curve fitting wavenumbers of subpeaks of asphalt mastic in the 1780–900 cm^−1^ range.

Numbers	1	2	3	4	5	6	7	8	9	10
Base mastics	1645	1600	1533	1460	1420	1377	1300	1161	1080	1032
SBS-modifiedmastics	1645	1600	1533	1460	1420	1377	1300	1161	1032	967

**Table 6 materials-12-02627-t006:** Functional group indexes of the mastics.

Samples		Aromatic Index	Branched Aliphatic Index	Sulfoxide Index	Long Chain Alkane Index	Butadiene Index
Base mastics	0–0	0.0284	0.5904	0.0008	0.0692	-
	8–8	0.0300	0.5169	0.0068	0.0483	-
	15–8	0.0302	0.3629	0.0145	0.0379	-
	25–8	0.0317	0.4119	0.0382	0.0281	-
SBS-modified mastics	0–0	0.0393	0.2840	0.0006	0.0641	0.0208
	8–8	0.0230	0.2756	0.0034	0.0356	0.0119
	15–8	0.0296	0.2072	0.0107	0.0318	0.0089
	25–8	0.0349	0.2287	0.0246	0.0258	0.0052

**Table 7 materials-12-02627-t007:** Surface height of the samples under different test conditions (nm).

Test Conditions	0–0	8–8	15–8	25–8
Base mastics	128.56	96.69	78.34	59.41
SBS-modified mastics	80	51.37	49.87	43.64

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
