# Peer review of "Investigation of the Microcharacteristics of Asphalt Mastics under Dry–Wet and Freeze–Thaw Cycles in a Coastal Salt Environment"

_materials, 2019, doi:10.3390/ma12162627_

Round 1
Reviewer 1 Report
1. Please explain about the mechanism of deterioration or aging of the asphalt material due to the accumulation of chloride ions. In concrete pavements, deicer salts (or chloride ions) deteriorate the microstructure of concretes and affect the hydration products in various ways. But, I am not sure how chloride ions can affect the performance or properties of asphalt pavements. You have provided two references (i.e., 3 and 4) to explain this mechanism. But, these two references are in Chinese and the reviewer is unable to judge the hypothesis used in your study. Please further explain the deterioration mechanism and provide English references.
2. In Table 2, please use English terms.
3. Please explain why you used SBS-modified binder in your study. I understand that this could be due the common use of SBS-modified in many regions. Please mention it if this was your motivation to use SBS-modified asphalt and provide reference(s) for that. You may want to use the following references:
· https://doi.org/10.1016/j.eurpolymj.2018.10.049
· https://doi.org/10.1016/j.conbuildmat.2017.06.115
· https://doi.org/10.1016/j.conbuildmat.2011.03.005
· https://doi.org/10.1016/j.fuel.2011.05.021
4. It would also be helpful if you could add the concentration of SBS in SBS-modified binder.
5. Fig. 6 has two parts. Please elaborate on which part shows what.
6. “Compared with the spectra of the base and the SBS-modified asphalt binder, the SBS-modified asphalt binder…”. What does this mean?
7. How do you use FTIR information to discuss the effects of FT cycles on the properties of the asphalt binders? There is no discussion on this!
Author Response
Dear Editors and Reviewers:
We thank you sincerely for commenting on our manuscript once again entitled “Investigation of the micro-characteristics of asphalt mastics under dry-wet and freeze-thaw cycles in coastal salt environment” (ID:543193). We have made modifications according to the reviewers’ comments in the revised manuscript. The modifications to the manuscript are highlighted in pink. The point-to-point responses to the reviewers’ comments are listed below.
Point 1: Please explain about the mechanism of deterioration or aging of the asphalt material due to the accumulation of chloride ions. In concrete pavements, deicer salts (or chloride ions) deteriorate the microstructure of concretes and affect the hydration products in various ways. But, I am not sure how chloride ions can affect the performance or properties of asphalt pavements. You have provided two references (i.e., 3 and 4) to explain this mechanism. But, these two references are in Chinese and the reviewer is unable to judge the hypothesis used in your study. Please further explain the deterioration mechanism and provide English references.
Response 1: We thank the reviewer for pointing out these problems. We had added the corresponding literatures on the deterioration mechanism of chloride ion on pavement performance in paper, which was also listed here.
Obika et al. [1] conduct a research about the damages produced by the salt that reaches the bituminous mixtures in hot climates, determining that the salt crystallization, solubility and crystal pressures have important roles in the damage produced in the mixture.
Luis et al. [2] research the mechanical properties of three types of bituminous mixtures affect by salt (sodium chloride, NaCl), and observe that the hot mix asphalt is scarcely affected when it is submerged in salt water, neither when salt is added as aggregate. However, the porous mixture is more susceptible to the effect of salt for all salt interaction, especially when the aggregate has been submerged in salt water before making the bituminous mixture. The salt in the porous mixtures has a harmful effect, it causes a loss of internal friction.
Starck et al. [3] insisted that the deicing agent decreased the stiffness and risen the softening point of the binder, viscosity of deicing-agent binders had risen after immersion in deicing agent solutions. This indicated the binder was aged during the F-T cycles and salt solution soaking process.
[1] Obika, B., Freer-Hewish, R.J., et al. Soluble salt damage to thin bituminous road and runway surfaces. Q. J. Eng. Geol. Hydrogeol. 1989, 22, 59–73.
[2] Luis Juli-Gándara, Ángel Vega-Zamanillo, Miguel Á., et al. Sodium chloride effect in the mechanical properties of the bituminous mixtures. Cold Regions Science and Technology. 2019, 164, 1-6.
[3] P. Starck, B. Lofgren. Influence of de-icing agents on the viscoelastic properties of asphalt mastics, J. Mater. Sci. 2007, 42 (2) 676–685.
Point 2: In Table 2, please use English terms.
Response 2: We are very sorry for our incorrect writing. Chinese words have been translated into English.
Point 3: Please explain why you used SBS-modified binder in your study. I understand that this could be due the common use of SBS-modified in many regions. Please mention it if this was your motivation to use SBS-modified asphalt and provide reference(s) for that.
Response 3: We thank the reviewer for pointing out these problems.
Currently, the most commonly used polymer for bitumen modification is the styrene–butadiene–styrene (SBS) followed by other polymers such as ethylene vinyl acetate EVA, styrene butadiene rubber (SBR) and polyethylene [1,2]. SBS block copolymers are classified as elastomers that increase the elasticity of bitumen and they are probably the most appropriate polymers for bitumen modification by improving the temperature susceptibility of binder [3–5].
[1] Airey GD. Styrene butadiene styrene polymer modification of road bitumen. J. Mater Sci.. 2004, 99: 951–99.
[2] Sengoz, B., & Isikyakar, G. Evaluation of the properties and microstructure of SBS and EVA polymer modified bitumen. Construction and Building Materials. 2008, 22, 1897–1905.
[3] Valtorta D, Poulikakos LD, Partl MN, Mazza E. Rheological properties of polymer modified bitumen from long-term field tests. Fuel, 2007, 86: 938–48.
[4] Wen G, Zhang Y, Sun K, et al. Rheological characterization of storage-stable SBS-modified asphalts. Polym. Test. 2002, 21:295–302.
[5] Wu S, Pang L, Mo L, et al. Influence of aging on the evolution of structure, morphology and rheology of base and SBS modified bitumen. Constr. Build Mater, 2009, 23: 1005–10.
Point 4: It would also be helpful if you could add the concentration of SBS in SBS-modified binder.
Response 4: We thank the reviewer for pointing out these problems.
In this paper, the SBS modified asphalt sample we selected was commercial asphalt, which were supplied by Sinopec Zhenhai Refining & Chemical Company (Ningbo, Zhejiang Province, China). The concentration of SBS polymer was 4.5 wt.% of the asphalt binders in this paper, which is the best concentration in accordance with the standard of China’s specification (JTG F40-2004). In follow-up research work, we will study the effect of SBS concentration on the chemical properties of the SBS modified asphalt.
Point 5: Fig. 6 has two parts. Please elaborate on which part shows what.
Response 5: In Fig. 6, a label has been added under each part for explanation.
Point 6: “Compared with the spectra of the base and the SBS-modified asphalt binder, the SBS-modified asphalt binder…”. What does this mean?
Response 6: We thank the reviewer for pointing out these problems.
This statement has been corrected that “Compared with the infrared spectrum of the base binder, the spectrum of the SBS-modified binder includes not only all the peaks of the base binder but also a special peak at 967 cm-1, which corresponds to the butadiene double bonds (—CH=CH—).”
Point 7: How do you use FTIR information to discuss the effects of FT cycles on the properties of the asphalt binders? There is no discussion on this!
Response 7: We thank the reviewer for pointing out these problems.
Yes, in this paper, the salt environment “DW-FT” cycles test is to simulate the early damage caused by the high temperature and high humidity in summer and the micro-freezing-thawing in winter of the natural environment of the saline area of the southern coastal environment in China. The focus of this paper is the coupling effect of “DW” and “FT” on the aging of the asphalt mastics, so this paper aims to examine the salt “DW-FT” cycles action, the aging of the asphalt mastics was not studied separately by the effect of the“DW”cycle or the “FT” cycle.
Many grammatical or typographical errors have been revised. All the lines and pages indicated above are in the revised manuscript. Thank you and all the reviewers for the kind advice.
With kindest regards,
Yours Sincerely
Qin ling, ZHANG
07/16/2019

Reviewer 2 Report
While the paper is sound in terms of the experiments, observations, and conclusions, there are a few points to be satisfied before the paper is ready for publication.
1. Table 2 has some words in Chinese. Please correct this.
2. English usage and spelling should be improved.
3. This reviewer think it will be useful if the authors provide some additional information on the Semi-quantitative analysis used in this study.
4. To enrich the conclusion, and to shed light on future researches, current theoretical models of asphalt concrete (which are granular materials in nature) can be cited here with the aim of: “The results of this paper can accompany the existing theoretical models describing asphalt concrete and granular media, as presented in the following papers, for design and analysis purposes.”
Nima Nejadsadeghi, Luca Placidi, Maurizio Romeo, Anil Misra, Frequency band gaps in dielectric granular metamaterials modulated by electric field, Mech. Res. Commun. 2019, 95, 96-103.
Anil Misra, Nima Nejadsadeghi, Longitudinal and transverse elastic waves in 1D granular materials modeled as micromorphic continua, Wave Motion, 2019, 90, 175-195.
5. The manuscript needs better description of the mechanical properties of asphaltic materials in pavement design, such as linear and nonlinear viscoelastic properties. They may use available literature such as the following reference:
Darabi, M. K., Huang, C. W., Bazzaz, M., Masad, E. A., & Little, D. N. (2019). Characterization and validation of the nonlinear viscoelastic-viscoplastic with hardening-relaxation constitutive relationship for asphalt mixtures. Construction and Building Materials, 216, 648-660.
Bazzaz, M. Experimental and Analytical Procedures to Characterize Mechanical Properties of Asphalt Concrete Materials for Airfield Pavement Applications. In Civil, Environmental and Architectural Engineering, Ph.D., University of Kansas, Lawrence, KS, 2018. p. 247.
This paper will recommend for publication if the authors consider the above suggestions to improve the quality of the manuscript. Some editing is still needed. I believe it will be done before publishing.
Author Response
Dear Editors and Reviewers:
We thank you sincerely for commenting on our manuscript once again entitled “Investigation of the micro-characteristics of asphalt mastics under dry-wet and freeze-thaw cycles in coastal salt environment” (ID:543193). We have made modifications according to the reviewers’ comments in the revised manuscript. The modifications to the manuscript are highlighted in pink. The point-to-point responses to the reviewers’ comments are listed below.
Point 1: Table 2 has some words in Chinese. Please correct this.
Response 1: Chinese words have been translated into English.
Point 2: English usage and spelling should be improved.
Response 2: We are very sorry for our incorrect writing. The spelling and syntax errors have been checked and corrected.
Point 3: This reviewer think it will be useful if the authors provide some additional information on the Semi-quantitative analysis used in this study.
Response 3: Thank you for pointing this out.
Semi-quantitative analysis is a compromise and is usually taken when it is very difficult to achieve quantitative analysis. Since the observation sample of the infrared spectrum in this paper is made by a KBr pellets technique[1], there are many uncontrollable factors in the preparation and testing of the sample, which makes the spectrum image not very accurate. Therefore, semi-quantitative analysis can be used according to the infrared spectrum. This method is used in the literature[2,3].
[1] Mansour E . Semi-quantitative analysis for FTIR spectra of Al2O3-PbO-B2O3-SiO2 glasses. Journal of Non-Crystalline Solids, 2012, 358 (3): 0-460.
[2] Kotyczka-Morańska M. Semi-quantitative and multivariate analysis of the thermal degradation of carbon-oxygen double bonds in biomass. Journal of the Energy Institute, 2018: S1743967117307882-.
[3] Xinyu Zhang, Lin Zhang, Xiang Zou, et al. Semi-quantitative analysis of microbial production of oxalic acid by montmorillonite sorption and ATR-IR. Applied Clay Science, 2018, (162): 518-523.
Point 4: To enrich the conclusion, and to shed light on future researches, current theoretical models of asphalt concrete (which are granular materials in nature) can be cited here with the aim of: “The results of this paper can accompany the existing theoretical models describing asphalt concrete and granular media, as presented in the following papers, for design and analysis purposes.”
Nima Nejadsadeghi, Luca Placidi, Maurizio Romeo, Anil Misra, Frequency band gaps in dielectric granular metamaterials modulated by electric field, Mech. Res. Commun. 2019, 95, 96-103.
Anil Misra, Nima Nejadsadeghi, Longitudinal and transverse elastic waves in 1D granular materials modeled as micromorphic continua, Wave Motion, 2019, 90, 175-195.
Response 4: We thank the reviewer for pointing out these problems. We have carefully reviewed the references provided by the reviewers and read some of the relevant literature. In the follow-up research work, we will carry out the theoretical model of asphalt concrete and will presented in the following papers.
Point 5. The manuscript needs better description of the mechanical properties of asphaltic materials in pavement design, such as linear and nonlinear viscoelastic properties.
Response 5: Thank you for suggestion.
In follow-up research work, the mechanical properties of asphalt mastics would be investigated by using dynamic shear rheometer (DSR) before and after different numbers of “DW-FT” cycles. the rheological properties of asphalt mastics in the linear and nonlinear viscoelastic region will be investigated by the multiple stress creep recovery (MSCR) test and the linear amplitude sweep (LAS).
Many grammatical or typographical errors have been revised. All the lines and pages indicated above are in the revised manuscript. Thank you and all the reviewers for the kind advice.
With kindest regards,
Yours Sincerely
Qin ling, ZHANG
07/16/2019

Reviewer 3 Report
Review: materials-543193 - Review
Investigation of the micro-characteristics of asphalt mastics under dry-wet and freeze-thaw cycles in coastal salt environment.
Abstract:
All components present;
Data are accurate and match text; Well‐written, concise, clear; Subject matter is original, press worthy, of major general interest.1.Introduction:
Well‐written, concise;
Hypothesis and purpose of study are clearly and concisely presented; Data are accurate; hypotheses are correctly presented and fully supported by text; Current references that will be of interest to readers.Remark:
??, ??, ?? and ???% indexes were obtained by the GPC test
Abbreviation need to be written fully when the appear the first time in your text.
2.Materials and methods:
All research components are present, clearly and stated. Procedures are clear, concise, and easily replicable;Although, some small remarks listed below:
Standard test methods of bitumen and bituminous mixtures for highway engineering. JTG E20-2011, Beijing, China, 2011. (in Chinese)
Can’t you refer to international standards?
2.1. Material Properties
The base asphalt binder (AH-70)
I wonder if you have data concerning the wax content? Should be nice to be added.
Table 1. Properties of asphalt binders.
General remark for the whole paper:
The accuracy of your presented data is missing. “standard deviation”
What is the accuracy of the measurements during your tests, mass, time, temperature…?
3.Results and discussion:
Overall this section is logically presented. It Summarizes the most important observations. Although statistical significance of findings is missing.
Some studies have found (although some disagreement exists) that the use of SBS in the modification of asphalt improves the flexibility at low temperatures and thereby enhances the cold weather performance of the modified asphalt [60]
Remark:
Can you comment on the effect of the wax in your base asphalt and the effect on your experiments and measurement?
Edwards Y., Redelius P., “Rheological effects of waxes in bitumen”, Energy & Fuels,
vol. 17, pp. 511-520, (2003).
Merusi F, Giuliani F (2011) Rheological characterization of
wax-modified asphalt binders at high service temperatures.
Mater Struct 44(10):1809–1820. https://doi.org/10.1617/
s11527-011-9739-4
YLVA EDWARDS, Influence of Waxes on Bitumen and Asphalt Concrete Mixture Performance, KTH Sweden
Figure 6. Spectra of asphalt binder, filler and mastic.
Remark: Try to improve de readability.
Figure 13. AFM topographic images of the mastics (20 μm × 20 μm).
Remark: Try to improve de readability.
Conclusions Statements and conclusions are clearly supported by data and are linked to goals. Study clearly advances knowledge. Study’s implications and limitations are completely and succinctly presented.Nevertheless, minor revisions of this text and the figures will improve the overall quality.
Author Response
Dear Editors and Reviewers:
We thank you sincerely for commenting on our manuscript once again entitled “Investigation of the micro-characteristics of asphalt mastics under dry-wet and freeze-thaw cycles in coastal salt environment” (ID:543193). We have made modifications according to the reviewers’ comments in the revised manuscript. The modifications to the manuscript are highlighted in red. The point-to-point responses to the reviewers’ comments are listed below.
Introduction
??, ??, ?? and ???% indexes were obtained by the GPC test. Abbreviation need to be written fully when they appear the first time in your text.
Response: Thank you for suggestion.
The abbreviation of ??, ??, ?? and ??? had been written fully when they appear the first time in my text. The modified content has been marked in blue.
Materials and methods
Standard test methods of bitumen and bituminous mixtures for highway engineering. JTG E20-2011, Beijing, China, 2011. (in Chinese)
Can’t you refer to international standards?
Response: We thank the reviewer for pointing out these problems.
In this paper, the research focus is to reveal the deterioration mechanism of the performance of asphalt pavement surface materials in the coastal areas of southern China, such as salt and climate, based on the standards of Chinese highway design and construction. It is designed for China's specific regional environment, traffic conditions and other conditions, but also formed by reference and drawing on international standards.
2.1. Material Properties
The base asphalt binder (AH-70), I wonder if you have data concerning the wax content ? Should be nice to be added.
Response: Thank you for suggestion.
The wax content in the base asphalt binder (AH-70) has been added in the Table 1.
Results and discussion:
Can you comment on the effect of the wax in your base asphalt and the effect on your experiments and measurement?
Response: Thank you for suggestion.
We have carefully reviewed the references provided by the reviewers and read some of the relevant literature. In the follow-up research work, we will study the influence of different kinds of wax and wax content on binder properties, which will be evaluated using different types of laboratory equipment before and after different numbers of “DW-FT” cycles, such as dynamic shear rheometer (DSR), bending beam rheometer (BBR), differential scanning calorimeter (DSC), fourier-transform infrared (FTIR) spectroscopy, gel permeation chromatography (GPC) and atomic force microscopy (AFM) and so on.
In general, Figure 6 and Figure 13 already were edited in order to improve readability. The accuracy of the experimental data in this paper was checked again. Many grammatical or typographical errors have been revised.
With kindest regards,
Yours Sincerely
Qin ling, ZHANG
08/07/2019
